# Optimal path planning of Unmanned Aerial Vehicles (UAVs) for targets touring: Geometric and arc parameterization approaches

Mohammad Forkan[1,2], Mohammed Mustafa Rizvi[1]*, Mohammad Abul Mansur Chowdhury[2]

**1** Department of Mathematics,University of Chittagong, Chittagong, Bangladesh, **2** JNIRCMPS, University of Chittagong, Chittagong, Bangladesh

☯ These authors contributed equally to this work.
* mmrizvi@cu.ac.bd

**Data Availability Statement:** Data has not been used.

**Funding:** M. Forkan received grants from Ministry of National Science & Technology, Bangladesh for

## Abstract

The path planning problem for unmanned aerial vehicles (UAVs) is important for scheduling the UAV missions. This paper presents an optimal path planning model for UAV to control its direction during target touring, where UAV and target are at the same altitude. Geometric interpretation of the given model is provided when the vehicles consider connecting an initial position to the destination position with specific target touring. We develop a nonlinear constrained model based on an arc parameterization approach to determine the UAV's optimal path touring a target. The model is then extended to touring finite numbers of targets and optimizing the routes. The model is found reliable through several simulations. Numerical experiments are conducted and we have shown that the UAV's generated path satisfies vehicle dynamics constraints, tours the targets, and arrives at its destination.

## 1 Introduction

In vehicle and robotic path planning, the shortest path calculation between any two given configurations is of fundamental interest. Due to the technology development, the usage of autonomous air vehicles is rising in both the military and civic. That is why the need of path planning becomes essential which is the main motivation of this paper. In this view many researchers have been trying to prescribed a model to find the shortest distance. Dubin [1] determined the smoothest and shortest path for vehicles confined to forward motion and unidirectional turns (left-turn or right-turn) from fixed initial to final configurations in the 2D plane. The author defined a *CSC* or a *CCC* path, or a subset thereof, as the minimum path that meets the maximum-curvature bound between two points with specific orientations,where *C* represents circular arc and *S* represents straight-line tangent to *C*; that is, the path will begin and end with a turn and its middle segment could be straight or turn. According to Dubins', the shortest path will be one of the six combinations *LSL, RSL, RSR, LSR, LRL,* and *RLR*, or a degenerate instance of these, where *L* and *R* represent clockwise (left-turn) and

providing financial help in the form of NST fellowship with Reference no. 120005100-3821117, Reg. no. 9 & Session: 2020-2021.

**Competing interests:** The authors have declared that no competing interests exist.

counter-clockwise (right-turn) circular arcs, respectively. When one or more of the segments are of zero length, this is referred to as a degenerate case.

A variation of a Dubins' path that allows backward motion is studied by Reeds and Shepp [2]. Sussmann and Tang [3] and Boissonnat et al [4] solved the Reed's problem [2] using the Pontryagin maximum principle (PMP) [5]. The PMP provides a necessary condition for a trajectory to be optimal. Moreover, Shkel and Lumelsky [6] made classification for the Dubins set, while Chitsaz and Lavalle [7] discussed the directions of the paths, which are more efficient approaches to calculate the shortest path. Recently, Kaya [8] have studied the reformulation of Dubins path based on optimality principle as a time-optimal collision-free trajectory of Dubins vehicle that is followed in this paper. Later, the author extended this approach to the problem of computing Dubins' interpolating curves, where a shortest curvature-constrained path through a given sequence of points [9]. Dubins path and its variants have been widely analyzed for optimal path planning of UAVs [10–12] and robots [13]. Since the past decade, there has been a significant increase in interest towards UAVs. It has increasingly been used in a wide range of applications such as logistics and surveillance [14], aerial forest fire detection [15], target observation [16], traffic monitoring and management [17], online commerce [18], geographic monitoring [19], scientific data collection [20], humanitarian relief [21], and disaster assessment and response [22–24].

The Dubins' routes have a number of findings in the literature [25–30]. The investigation of the accessibility zones of Dubins' routes and the Dubins' synthesis problem are described in [25, 26]. The three-point Dubins' issue is described as an extension of the Dubins' path planning problem [27, 28]. The three point Dubins' problem is to determine the curvature constrained path between initial and final points, such that the path passes through the third point or a fixed target. Another generalization is the Dubins' interval problem that is presented in [29, 30]. According to the studies [31–33], Dubins' path converging tangentially to a specified line and tangent to a target circle. Aside from these results, various studies have also been conducted on the potential uses of Dubins' routes, such as the difficulty of chasing a moving target [34–36].

The present paper proposes geometric and mathematical models in Sections 2 and 4 to approximate the optimal path from the origin to the destination and intercepting a finite number of targets. The proposed models also fit any UAV applications to pursue and intercept the targets following a minimum path length or minimum time. It is expected that the proposed models should have the following practical attributes:

1. The method should optimize the heading angles to determine the UAVs route to generate the shortest path of the UAV's flight through intercepting targets. This is an important attribute as this ensures that using those heading angles, one can optimize the path or time for UAV tour without evaluating each of the feasible path individually. However, as far as we know, most popular methods characterize the optimal path by finding the length of each possible feasible path using algorithms based on Markov-Dubin concepts. Therefore, our concern here is to introduce a mathematical model and develop its capability of approximating the optimal trajectory in terms of length and time by solving a mathematical problem.

2. The method should generate an optimal path with least possible computational time. This attribute has important practical consequences because finding the length of each of the feasible paths can itself be very costly. Thus, we are keen to know the advantages of the proposed models to generate the optimal path with minimal computational effort.

3. The method should generate the optimal trajectory of UAV tour when UAV is scheduled to tour through multiple number of targets. This attribute has critical importance as in many real-life applications we often intend to use UAV to cover several targets where human access is harmful or human effort is non-feasible. In such a circumstance, it is important to introduce a model which can tour many targets through an optimized trajectory. Hence, we are interested to see the capabilities of the proposed method to approximate optimal trajectory with the task of touring numerous targets.

Our proposed method efficiently satisfies all three attributes. The strengths and advantages of the proposed method with Attributes (i)-(iii) are illustrated through the examples (see Section 5), and the results are demonstrated in Figs 2–4. We show that the efficient performance obtained by the proposed technique successfully generates the shortest time and the shortest path for multiple targets.

The rest of this paper comprises the following sections. The geometrical interpretation of optimal path planning of UAV for a target touring is described in Section 2. In Section 3, we illustrate reformulation of Dubins' problem as a time-optimal control problem and Pontryagin's Maximum Principle. In Section 4.1, the model for finding a optimal path of UAV to tour a single target is proposed and the extended model for optimal path of UAV to intercept the multiple targets is derived in Section 4.2. Numerical experiments are conducted and we provide the results and discussions based on numerical experiments in Section 5. The last section presents the conclusion of the paper.

## 2 Geometrical interpretation path planning for single target touring

This section illustrates path planning for UAVs that start from an initial location, pass through a known target, and reach the finishing point. The proposed geometric approach describes an essential criterion for shortest route planning subject to the conditions that UAVs tour the target. Our proposed geometrical interpretation of a feasible optimal path of type *CSCSC* for a single target is the concatenation of two sub-paths of type *CS* and *CSC*. These sub-paths characterized by Dubins' path (Theorem 1, [8]) are demonstrated in Fig 1 in 2*D* environment. The complete methodology of constructing the paths for a UAV with a single target touring is given below.

In Fig 1, there is a target fixed at *T*. Let UAV's starting and finishing points are $P_I$ and $P_f$, respectively. $X_i$, $i = 1, \ldots, 8$ are defined to calculate *CS* and *CSC* curves. For instance, the curve *CS* is calculated adding arc $P_I X_1$ and straight line $X_1 T$ if UAV turns left, and $P_I X_2 + X_2 T$ if it turns right in 2D environment. The equations of Step-I describe how to approximate *CS* paths. The curve *CSC* is constructed as either any of paths presented in Step-II.

It is noted that initial configuration (a configuration consists of a position and a heading angle) $P_I(x_0, y_0, \psi_0)$ and the final configuration is $P_f(x_f, y_f, \psi_f)$. The fixed target location is $T(x_T, y_T, \psi_T)$, which is known in advance where $\psi_T$ is the angle at *T* formed by straight lines $X_1 T$ or $X_2 T$ presented in (1). According to the Fig 1, UAV moves from $P_I$ with a given heading angle $\psi_0$ and then flies towards target T, and after touring T, it turns along its destination $P_f$.

We divide the minimum path design for target touring problems into following two steps, as shown in Fig 1.

Step I: The first part of the optimal path includes two alternative feasible paths of type *CS*,

$$CS \equiv \begin{cases} LS \equiv \text{arc } P_I X_1 + \text{st. line } X_1 T, & \text{or,} \\ RS \equiv \text{arc } P_I X_2 + \text{st. line } X_2 T, \end{cases} \tag{1}$$

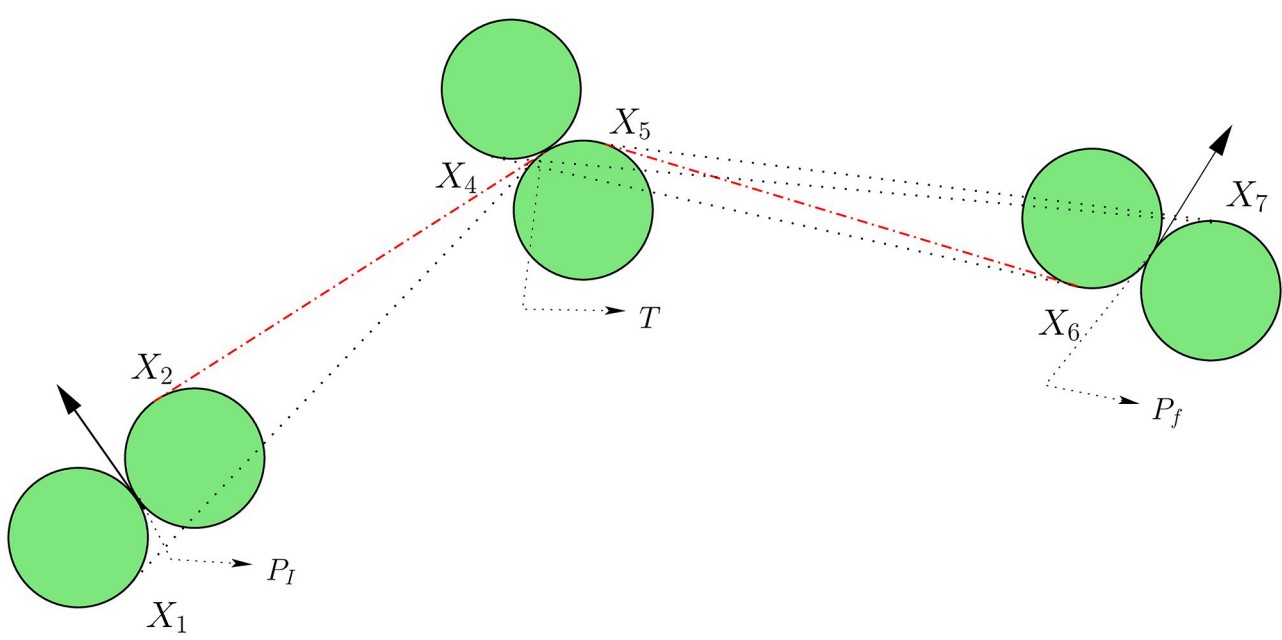

**Fig 1. Feasible optimal paths of UAV for single target touring.**

Step II: The UAV turns and moves towards its destination $P_f$ after completing the first path $CS$ through target $T$. This creates four possible feasible paths of type $CSC$ which are as follows:

$$CSC \equiv \begin{cases} LSL \equiv \text{arc } TX_4 + \text{st. line } X_4X_6 + \text{arc } X_6P_f, & \text{or,} \\ LSR \equiv \text{arc } TX_4 + \text{st. line } X_4X_7 + \text{arc } X_7P_f, & \text{or,} \\ RSL \equiv \text{arc } TX_5 + \text{st. line } X_5X_6 + \text{arc } X_6P_f, & \text{or,} \\ RSR \equiv \text{arc } TX_5 + \text{st. line } X_5X_7 + \text{arc } X_7P_f. \end{cases} \quad (2)$$

Similarly, it is easy to find the optimal path of UAV for $n$ targets touring. In that case, the optimal trajectory is composed of $(n + 1)$ sub-paths, with $2n$ alternative feasible sub-paths of type $CS$ and 4 alternative feasible sub-paths of type $CSC$. Note that, these feasible sub-paths hold the positions $X_i$ at consecutive times $t_i$, respectively, where $i = 0, 1, 2, \ldots, 3n+5$, $X_0 = P_I$ and $X_{3n+5} = P_f$. For an instance, when UAV tour two targets i.e. n = 2, then the optimal path is a concatenation of three curves, in which two curves are of 4 alternative feasible sub-paths of type $CS$ and one of 4 alternative sub-paths of type $CSC$.

## 3 Optimal control problem formulation and Pontryagin's maximum principle

In this section, we formulate Dubins' problem as an optimal control problem that minimizes the flyable path when UAV is set to fly from a fixed point to a destination through touring targets. It is expected that during the tour UAV must record the essential information about the target whose position is known in advance, and then UAV moves towards its known destination.

Our analysis assumes the UAV is moving with constant speed and subject to minimum turning radius constraint. The Dubins' problem can be constructed as to find a curve $\gamma(s)$ : $[t_0, t_f] \to \mathbb{R}^2$ through the prescribed points $P_I$ and $P_f$ in $\mathbb{R}^2$ at the time $t_0$ and $t_f$ respectively, such that (for details see, in [37])

1. $\gamma(s)$ is a continuously differentiable curve, parameterized with respect to its arc-length.

2. $\|\dot{\gamma}\| = 1$ and satisfies the Lipsehtiz condition, $\|\dot{\gamma}(s_2) - \dot{\gamma}(s_1)\| \leq \alpha s_2 - s_1$ for all $s_1, s_2$ in the domain of $\gamma(s)$.

3. $\gamma(s)$ satisfies the boundary conditions: $\gamma(t_0) = P_I, \dot{\gamma}(t_0) = v_0, \gamma(t_f) = P_f, \dot{\gamma}\left(t_f\right) = v_f$, where $v_0$ and $v_f$ are velocity vectors at $t_0$ and $t_f$ respectively.

4. $\gamma(s)$ has shortest length i.e., $t_f$ is minimal, if $\gamma(s)$ satisfies (i)-(iii).

Since Lipschitz functions are differentiable almost everywhere (a.e.) by (ii), it follows that $\gamma$ has a curvature a.e and that the curvature is upper bounded by $\alpha$, is defined as the length of its acceleration vector; i.e.,

$$\kappa(t) = ||\ddot{\gamma}(t)||, \text{with } ||\dot{\gamma}(t)|| = 1, \tag{3}$$

where $\kappa(t) \leq \alpha$, for almost all $t \in [t_0, t_f]$. The motion of a vehicle in the plane is $1/\kappa(t)$, which represents the instantaneous turning radius and in the Dubins' problem, the turning radius is constrained to be at least $1/\alpha$. Thus, the shortest path of UAV can be written as minimization of the arc-length functional

$$\int_{t_0}^{t_f} ||\dot{\gamma}(t)||\, dt = t_f - t_0, \tag{4}$$

where $||\dot{\gamma}(t)|| = 1$ from (3) is used, for a.e. $t \in [t_0, t_f]$.

We note that, since objective function of the problem does not depend apparently on the time parameter $t$, the value of initial time may be selected arbitrary, hence there is no loss of generality to choose $t_0 = 0$. Therefore the Dubins' problem can be represented as (P): Minimize (4), subject to the conditions are $\gamma(0) = P_I, \gamma(t_f) = P_f$, which are initial and destination points, respectively, velocities at initial and end points are $\dot{\gamma}(0) = v_0, \dot{\gamma}(t_f) = v_f$, respectively. More-over,

$$||\ddot{\gamma}(t)|| \leq \alpha, ||\dot{\gamma}(t)|| = 1, \text{for a.e } t \in [0, t_f], \tag{5}$$

where $||v_0|| = ||v_f|| = 1$.

Using (3)–(5), we now are able to formulate Dubins' problem (P) as an optimal control problem (for more details can be seen in [8]). It is required to find the continuously differentiable optimal possible path between two fixed points in $\mathbb{R}^2$, with prescribed tangent vectors at the end points. Thus, as mentioned in (4), the quantity that we desire to minimize is

$$\int_0^{t_f} \sqrt{\dot{x}^2 + \dot{y}^2}\, dt = \int_0^{t_f} ||\dot{\gamma}(t)||\, dt, \tag{6}$$

where $(x(t), y(t)) \in \mathbb{R}^2$ is the planar position of the UAV and $\psi \in [0, 2\pi]$ is the heading angle, the velocity vector $\dot{\gamma}(t)$ of the curve $\gamma(t)$ makes with the horizontal. Taking the unit velocity vector and turn rate $\omega(t) = \dot{\psi}(t)$ as control variable, the equations of motion are

$\dot{x}(t) = \cos\psi(t)$ and $\dot{y}(t) = \sin\psi(t)$, and these verify that $||\dot{\gamma}(t)|| = 1$ in (6). Furthermore,

$$||\ddot{\gamma}||^2 = \ddot{x}^2 + \ddot{y}^2 = \dot{\psi}^2. \tag{7}$$

The relation (7) implies that, $|\dot{\psi}(t)|$ is the curvature. Since the quantity $\dot{\psi}(t)$ can be positive or negative, so it can be quoted to as the signed curvature. Consider pictorially the UAV travelling along a circular path (C). If $\dot{\psi}(t) > 0$ then the UAV travels in the counter-clockwise direction i.e. along left turns subarc (L) and if $\dot{\psi}(t) < 0$ then the UAV travels in the clockwise direction i.e. along right turns subarc (R), and also $\dot{\psi}(t) = 0$, if the UAV travels in the straight line (S).

Let us consider that the angles of the directions at the points $P_I$ and $P_f$ are $\psi_0$ and $\psi_f$, respectively and $\dot{\psi}(t) = \omega(t)$. Hence, Dubins' problem (P) can be stated as an optimal control problem.

$$\min \qquad t_f = \int_0^{t_f} dt$$

subject to

$$\dot{x}(t) = \cos\psi(t), \; x(0) = x_0, \; x(t_1) = x_1, \; \ldots, \; x(t_N) = x(t_f) = x_f,$$

$$\dot{y}(t) = \sin\psi(t), \; y(0) = y_0, \; y(t_1) = y_1, \; \ldots, \; y(t_N) = y(t_f) = y_f, \tag{8}$$

$$\dot{\psi}(t) = \omega(t), \; \psi(0) = \psi_0, \; \psi(t_1) = \psi_1, \; \ldots, \; \psi(t_N) = \psi(t_f) = \psi_f,$$

$$|\omega(t)| \le \alpha, \; \text{for a.e } t \in [0, t_f], \; N = 1, 2, \ldots, (3n + 5) \text{ and } n \in \mathbb{N},$$

where $x$, $y$ and $\psi$ are the state variables and $\omega$ is the control variable and $n$ is the number of targets.

## 3.1 Pontryagin's maximum principle

In this subsection, we will state the necessary conditions of optimality for Pontryagin's maximum principle.

Let us consider the costate of state variable $\bar{x} \in X$ by $\eta := [\eta_1, \eta_2, \eta_3]$ in cotangent space $T_{\bar{x}}^* X$. According to Pontryagin's maximum principle [5], if a trajectory $\bar{x}(.) = [x(.), y(.), \psi(.)]^T \in X$ associate with measurable control $\omega(.) \in [-\alpha, \alpha]$ on $[0, t_f]$ are the solution of Problem (8), there exists a scalar parameter $\eta_0$ and a continuous mapping $t \mapsto \eta(.) \in T\bar{x}(.)^* X$ on $[0, t_f]$, satisfying $[\eta(t), \eta_0] \neq 0$ for $t \in [0, t_f]$, such that, a.e on $[0, t_f]$, the followings equations hold,

$$\dot{\eta}_1 = -H_x(t), \; \dot{\eta}_2 = -H_y(t), \; \dot{\eta}_3 = -H_\psi(t), \; a.e. \; t \in [0, t_f], \tag{9}$$

$$H(\bar{x}(t), \eta(t), \omega(t)) = 0, \tag{10}$$

$$\omega(t) \in \arg\max_{||v|| \le \alpha} H(x(t), y(t), \psi(t), \eta_0, \eta_1(t), \eta_2(t), \eta_3(t), v), \; a.e \; t \in [0, t_f]. \tag{11}$$

where $H(\bar{x}, \eta, \omega) = \eta_1 \cos(\psi) + \eta_2 \sin(\psi) + \eta_3 \omega + \eta_0$ is the Hamiltonian.

The optimality conditions (9)–(11) can now be re-written more explicitly as follows.

$$\dot{\eta}_3(t) = \bar{\eta}_1 \sin \psi(t) - \bar{\eta}_2 \cos \psi(t), \ \forall \ t \in [t_{N-1}, t_N], \ N = 1, 2, ..., (3n + 5), \tag{12}$$

$$\eta_0 + \bar{\eta}_1 \cos \psi(t) - \bar{\eta}_2 \sin \psi(t) + \eta_3(t)\omega(t) = 0, \ a.e. \ t \in [t_{N-1}, t_N), \ N = 1, 2, ..., (3n + 5), \tag{13}$$

$$\omega(t) = \begin{cases} \alpha & \text{if } \eta_3 > 0 \\ -\alpha & \text{if } \eta_3 < 0 \\ 0 & \text{if } \eta_3 = 0, \ a.e. \ t \in [t_{N-1}, t_N), \ N = 1, 2, ..., (3n + 5), \end{cases} \tag{14}$$

where $\bar{\eta}_1$ and $\bar{\eta}_2$ are real constants. It is here mentioned that the adjoint variable $\eta_3$ is nothing but the switching function for the control $\omega$.

## 4 Mathematical model for target touring

According to geometrical demonstration introduced in Section 2, we now propose a mathematical model to find the optimal path of the UAV for single target intercepting, and later extend this model for multiple targets touring is illustrated in Section 4.2.

### 4.1 Optimal path planning for single target touring

Under the premise that the feasible paths satisfying minimum turn radius criteria and intersect a target whose position is known in advance, we find an optimal path that consists of five sub-arc for a single target tour, which are either curve (*C*) or straight line (*S*). The required optimal path with a target touring can be classified as one of the following specific types as the Eqs (1) and (2) portray.

$$\{LSLSL, LSLSR, LSRSL, LSRSR, RSLSL, RSLSR, RSRSL, RSRSR\}. \tag{15}$$

Suppose the starting time $t_0 := 0$ and the terminal time $t_f := t_8$. Also set $l_i := t_i - t_{i-1}$, for $i = 1, 2, ..., 8$ that corresponds to the time duration or length for each subarc. We now solve the ODEs equations given in Problem (8) for $x(t), y(t), \psi(t)$ with the time interval $t_{i-1} \leq t \leq t_i$ yields,

$$\int_{t_{i-1}}^{t_i} \dot{x}(t) \, dt = \int_{t_{i-1}}^{t_i} \cos\psi(t) \, dt$$

$$\int_{t_{i-1}}^{t_i} \dot{y}(t) \, dt = \int_{t_{i-1}}^{t_i} \sin\psi(t) \, dt \tag{16}$$

$$\int_{t_{i-1}}^{t_i} \dot{\psi}(t) \, dt = \int_{t_{i-1}}^{t_i} \omega(t) \, dt$$

From (16), we obtain the position $(x(t_i), y(t_i))$ along turning curve (*C*) yields

$$x(t_i) = x(t_{i-1}) + (\sin\psi(t_i) - \sin\psi(t_{i-1}))/\omega(t),$$
$$y(t_i) = y(t_{i-1}) - (\cos\psi(t_i) - \cos\psi(t_{i-1}))/\omega(t), \tag{17}$$

also the position $(x(t_i), y(t_i))$ of UAV along straight line ($S$) can be defined as

$$
\begin{aligned}
x(t_i) &= x(t_{i-1}) + \cos\psi(t_i)(t_i - t_{i-1}), \\
y(t_i) &= y(t_{i-1}) + \sin\psi(t_i)(t_i - t_{i-1}),
\end{aligned}
\tag{18}
$$

and heading angles of UAV along the both curve ($C$) and straight line ($S$) can be obtained as

$$
\psi(t_i) = \psi(t_{i-1}) + \omega(t)(t_i - t_{i-1}),
\tag{19}
$$

where $\omega(t) = \alpha$, for left-turn circular arc; $\omega(t) = -\alpha$, for right-turn circular arc and $\omega(t) = 0$, for straight line.

Now we adopt mathematical approaches to construct the model for finding the optimal path of UAV with single target touring describe as follows.

**Algorithm (Shortest Path)**

**Step 1 (Input)**
　Set left-turn arc $\ell_i$, $i = 1, 4, 7$ right-turn arc $\ell_i$, $i = 2, 5, 8$, and st. line $\ell_i$, $i = 3, 6$.
　Assume $\ell_i \neq 0$, $i = 3, 6$.

**Step 2 (Determine the length from initial point to target point)**
　Set Curve-St.Line $:= CS$.
　Find $CS := \ell_1 + \ell_2 + \ell_3$ such that
　Turns left:
　$\ell_2 = 0$ and $CS := \ell_1 + \ell_3$,
　Or
　Turns right:
　$\ell_1 = 0$ and $CS := \ell_2 + \ell_3$.

**Step 3 (Determine the length from target point to finishing point)**
　Set Curve-St.Line-Curve $:= CSC$
　Find $CSC := \ell_4 + \ell_5 + \ell_6 + \ell_7 + \ell_8$ such that
　Turns left-st-left:
　$\ell_5 = 0$, $\ell_8 = 0$ and $CSC := \ell_4 + \ell_6 + \ell_7$,
　Or
　Turns left-st-right:
　$\ell_5 = 0$, $\ell_7 = 0$ and $CSC := \ell_4 + \ell_6 + \ell_8$,
　Or
　Turns right-st-left:
　$\ell_4 = 0$, $\ell_8 = 0$ and $CSC := \ell_5 + \ell_6 + \ell_7$,
　Or
　Turns right-st-right:
　$\ell_4 = 0$, $\ell_7 = 0$ and $CSC := \ell_5 + \ell_6 + \ell_8$.

**Step 4** Find optimal path that solves Problem (P) $:= \min(CS + CSC)$.

To determine optimal path for single target touring, we combine the feasible paths obtained in Steps 2 and 3 and thus optimal path be one of the feasible paths listed in (15). For instance, if we obtain optimal of type *RSLSR*, then we attained lengths with $l_1 = l_5 = l_7 = 0$ and $l_2, l_3, l_4, l_6, l_8 > 0$.

We now construct a model that gives the minimum path of UAV, when UAV moves from its origin to destination by touring a single target. The objective is to minimize the sum of lengths $l_i$, $i = 1, \ldots, 8$ which are the variables of the model. We further assume that the target position is at $(x_T, y_T, \psi_T) \equiv (x(t_3), y(t_3), \psi(t_3))$, starting point $(x_0, y_0, \psi_0) \equiv (x(t_0), y(t_0), \psi(t_0))$ and destination $(x_f, y_f, \psi_f) \equiv (x(t_8), y(t_8), \psi(t_8))$. According to Step I, we generate the expressions for $CS$ path that cover the time taken from starting point to $(x(t_3), y(t_3), \psi(t_3))$. Therefore, by putting i = 1,2 in (17) and $i = 3$ in (18) and then completing algebraic calculations, we obtain $CS$

path for the duration from $t_0$ to $t_3$ as below.

$$x(t_0) - x(t_3) + \frac{1}{\alpha}(-\sin\psi(t_0) + 2\sin\psi(t_1) - \sin\psi(t_2)) + l_3\cos\psi(t_2) = 0, \quad (20)$$

and

$$y(t_0) - y(t_3) + \frac{1}{\alpha}(\cos\psi(t_0) - 2\cos\psi(t_1) + \cos\psi(t_2)) + l_3\sin\psi(t_2) = 0. \quad (21)$$

where, $\psi(t_2) = \psi(t_3)$ as UAV follows straight line with preceding heading angle, when it leaves the turning circle.

Next, as stated in Step-II, we generate the equations for *CSC* path that cover the time taken from target intercepts point $(x(t_3), y(t_3), \psi(t_3))$ to destination $(x(t_8), y(t_8), \psi(t_8))$ of UAV. Thus, by putting $i = 4, 5, 7, 8$ in (17) and $i = 6$ in (18) and after algebraic calculations, we have path of type *CSC* for the duration from $t_3$ to $t_8$ as follows.

$$x(t_3) - x(t_8) + \frac{1}{\alpha}(-\sin\psi(t_2) + 2\sin\psi(t_4) - 2\sin\psi(t_5) + 2\sin\psi(t_7) - \sin\psi(t_8)) + l_6\cos\psi(t_5) = 0, (22)$$

and

$$y(t_3) - y(t_8) + \frac{1}{\alpha}(\cos\psi(t_2) - 2\cos\psi(t_4) + 2\cos\psi(t_5) - 2\cos\psi(t_7) + \cos\psi(t_f)) + l_6\sin\psi(t_5) = 0. (23)$$

where, $\psi(t_5) = \psi(t_6)$ as UAV follows straight line with preceding heading angle, when it leaves the target turning circle.

Thus, by combining (20)–(23), with the additional conditions $\sin\psi_f = \sin\psi_8$ and $\cos\psi_f = \cos\psi_8$, we propose the following model to obtain optimal path of UAV for a single target touring. Note that, we have replaced $\psi(t_k)$ by $\psi_k$, $k = 0, \ldots, 8$ for convenience of the notation.

$$\text{min} \qquad t_f = \sum_{i=1}^{8} l_i$$

subject to

$$x_0 - x_T + \frac{1}{\alpha}(-\sin\psi_0 + 2\sin\psi_1 - \sin\psi_2) + l_3\cos\psi_2 = 0,$$

$$y_0 - y_T + \frac{1}{\alpha}(\cos\psi_0 - 2\cos\psi_1 + \cos\psi_2) + l_3\sin\psi_2 = 0,$$

$$x_T - x_f + \frac{1}{\alpha}(-\sin\psi_2 + 2\sin\psi_4 - 2\sin\psi_5 + 2\sin\psi_7 - \sin\psi_8) + l_6\cos\psi_5 = 0, \quad (24)$$

$$y_T - y_f + \frac{1}{\alpha}(\cos\psi_2 - 2\cos\psi_4 + 2\cos\psi_5 - 2\cos\psi_7 + \cos\psi_8) + l_6\sin\psi_5 = 0,$$

$$\sin\psi_f - \sin\psi_8 = 0,$$

$$\cos\psi_f - \cos\psi_8 = 0,$$

$$l_i \geq 0, \quad i = 1, 2, ..., 8,$$

where

$$\psi_1 = \psi_0 + \alpha l_1, \psi_2 = \psi_1 - \alpha l_2, \psi_4 = \psi_2 + \alpha l_4, \psi_5 = \psi_4 - \alpha l_5, \psi_7 = \psi_5 + \alpha l_7, \psi_8 = \psi_7 - \alpha l_8. \tag{25}$$

## 4.2 Optimal path planning for *n* targets touring

We now establish a path planning model for *n* targets touring, where $n \in N$. We intend to minimize sum of lengths $l_i$, $i = 1, \ldots, 3n + 5$. We follow similar algebraic calculation that carried in (20) and (21), and obtain *n* number of *CS* type paths. The first *CS* type path includes the length between origin to 1st target intercepts. Following the second *CS* path starts from the 1st target considering this target point as origin and then follows the 1st-target turning circle, flies up to the 2nd target, and proceeds to generate the next path and so on. The length of last *CS* type path will be the length from $(n - 1)$th to *n*th target. These give us *n* pairs of equations for *x* and *y* variables and obtained equations are illustrated below.

Set, $\psi(t_k) = \psi_k$, $x(t_k) = x_k$, $k = 0, \ldots, 3n + 5$, and $\psi_{3j} = \psi_{3j-1}$, $j = 1, \ldots, n + 1$.

$$x_{3j-3} - x_{3j} + \frac{1}{\alpha}\left(-\sin\psi_{3j-3} + 2\sin\psi_{3j-2} - \sin\psi_{3j-1}\right) + l_{3j}\cos\psi_{3j} = 0, \; j = 1, \ldots, n, \tag{26}$$

and

$$y_{3j-3} - y_{3j} + \frac{1}{\alpha}\left(\cos\psi_{3j-3} - 2\cos\psi_{3j-2} + \cos\psi_{3j-1}\right) + l_{3j}\sin\psi_{3j} = 0, \; j = 1, \ldots, n, \tag{27}$$

The final path is the type of *CSC* and the length of this final curve is obtained from *n*th target to UAV's destination. As a result, we can formulate the equation for the *CSC* sub-path by doing a similar algebraic calculation persisted in (22) and (23), and the equations are described below.

$$\begin{aligned} x_{3n} - x_{3n+5} + \frac{1}{\alpha}\left(-\sin\psi_{3n} + 2\sin\psi_{3n+1} - 2\sin\psi_{3n+3} + 2\sin\psi_{3n+4} - \sin\psi_{3n+5}\right) \\ + l_{3n+3}\cos\psi_{3n+3} = 0, \end{aligned} \tag{28}$$

and

$$\begin{aligned} y_{3n} - y_{3n+5} + \frac{1}{\alpha}\left(\cos\psi_{3n} - 2\cos\psi_{3n+1} + 2\cos\psi_{3n+3} - 2\cos\psi_{3n+4} + \cos\psi_{3n+5}\right) \\ + l_{3n+3}\sin\psi_{3n+3} = 0, \end{aligned} \tag{29}$$

Hence, by combining (26)–(29) with additional auxiliary conditions, we can summarize the *n*

targets touring model as follows.

$$\min \qquad t_f = \sum_{i=1}^{3n+5} l_i, \quad n \in \mathbb{N}$$

subject to

$$\begin{cases} x_{3j-3} - x_{3j} + \dfrac{1}{\alpha}\left(-\sin\psi_{3j-3} + 2\sin\psi_{3j-2} - \sin\psi_{3j-1}\right) + l_{3j}\cos\psi_{3j} = 0, \\[2mm] y_{3j-3} - y_{3j} + \dfrac{1}{\alpha}\left(\cos\psi_{3j-3} - 2\cos\psi_{3j-2} + \cos\psi_{3j-1}\right) + l_{3j}\sin\psi_{3j} = 0, \end{cases}$$
$$j = 1, \ldots, n. \tag{30}$$

$$x_{3n} - x_{3n+5} + \frac{1}{\alpha}\left(-\sin\psi_{3n} + 2\sin\psi_{3n+1} - 2\sin\psi_{3n+3} + 2\sin\psi_{3n+4} - \sin\psi_{3n+5}\right) + l_{3n+3}\cos\psi_{3n+3} = 0,$$

$$y_{3n} - y_{3n+5} + \frac{1}{\alpha}\left(\cos\psi_{3n} - 2\cos\psi_{3n+1} + 2\cos\psi_{3n+3} - 2\cos\psi_{3n+4} + \cos\psi_{3n+5}\right) + l_{3n+3}\sin\psi_{3n+3} = 0,$$

$$\sin\psi_f - \sin\psi_{3n+5} = 0,$$

$$\cos\psi_f - \cos\psi_{3n+5} = 0,$$

$$l_i \geq 0, \text{ for } i = 1, \ldots, 3n+5.$$

where

$$\psi_j = \psi_{j-1} + \alpha_j l_j, \quad j = 1, \ldots, 3n+5. \tag{31}$$

$$\begin{cases} \alpha_j = \quad \alpha, \quad j \in \{1, 4, \ldots, (3n+4))\} \\ \alpha_j = -\alpha, \quad j \in \{2, 5, \ldots, (3n+5))\} \\ \alpha_j = \quad 0, \quad j \in \{3, 6, \ldots, 3(n+1)\} \end{cases}$$

## 5 Numerical experiments, results and analysis

The numerical experiments to determine the optimum UAV path for the preplanned target tour is presented in this section. We provide the initial and destination locations of UAV along with target positions into the algorithms. The task set for UAV is that UAV begins from a fixed starting point and tour (intersect) the target, then reach at its fixed destination point. To test the capabilities of the models (24) and (30), we set three experiments through Examples 1 and 2 as follows.

### Example 1

Set starting point $P_0(0, 0, 2\pi/3)$ and destination point $P_f(20, 8, \pi/6)$. The task is given to UAV to tour target $T(6, 10)$ with a minimum path chosen from (15). It is assumed that the minimum radius of the turning curve circle for UAV is $r = 2$, and UAV also follows the same radius circle when it intercepts the target and moves towards its destination $P_f$.

The results show that *RSRSL* path is the optimum path to accomplish the task, which is shown in Fig 2. The lengths obtained by the algorithms are demonstrated in Table 1. The total length of the path obtained by the algorithm is 26.51.

If we change the target location at (13, 0), the algorithm chooses *RSLSR* from (15) as a minimum path to complete the task. The optimal path is shown in Fig 3, and the length of the portions of the path can be seen in Table 1. In this case, The total length of the path obtained by

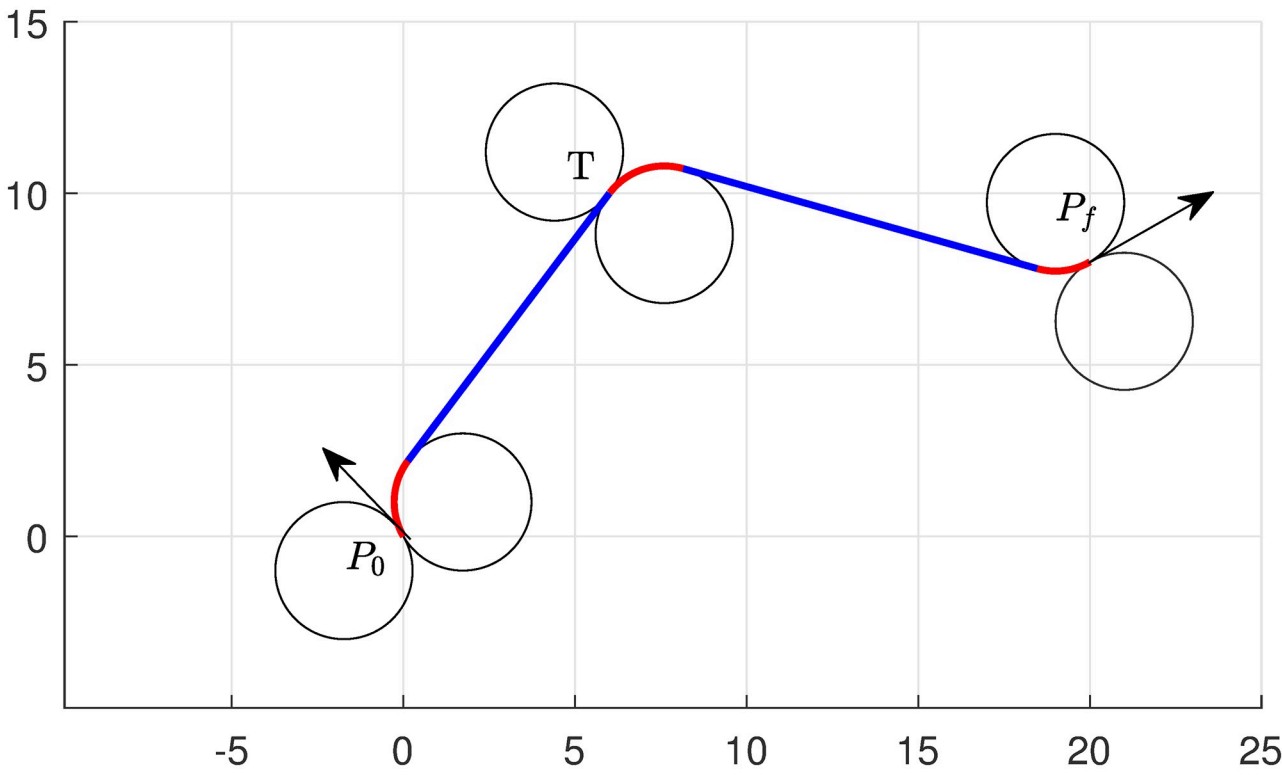

**Fig 2. Minimum path of UAV with single target touring of type RSRSL.**

the algorithm is 27.04. Note that the computational taken by the solver (KNITRO [38]) is 0.125 seconds, the total time elapsed by the system 0.155 seconds, and the simulation time taken by MATLAB is approximately 1.5 seconds.

### Example 2

We conduct the experiment for multiple targets, and set starting and destination points at $P_0(0, 0, \pi/2)$ and $P_f(30, 18, \pi/6)$, respectively. UAV will now tour targets $T_1(10, 10)$ and $T_2(25, 5)$, and we require to find the optimal path to complete the task.

We solve model (30), and the optimal path obtained by the algorithm is presented in Fig 4. The lengths correspond to the path is given in Table 2. We can see that the algorithm choose *RSRSLSR* path to obtain minimum path. The total length of the path obtained by the algorithm is 46.07. It is noted that the computational taken by the solver (Bonmin [35]) is 0.0625 seconds, the total time elapsed by the system 0.09 seconds, and the simulation time taken by MATLAB is approximately 2 seconds.

**Table 1. Single target touring—Numerical performance of model (24).**

| Targets/lengths turn | Left turn | Right turn | Straight turn | Left turn | Right turn | Straight turn | Left turn | Right turn |
|:---:|:---:|:---:|:---:|:---:|:---:|:---:|:---:|:---:|
| | $L_{l_1}$ | $L_{l_2}$ | $L_{l_3}$ | $L_{l_4}$ | $L_{l_5}$ | $L_{l_6}$ | $L_{l_7}$ | $L_{l_8}$ |
| $T(6, 10)$ | 0 | 2.35 | 9.76 | 0 | 2.40 | 10.40 | 1.60 | 0 |
| $T(13, 0)$ | 0 | 4.72 | 11.13 | 2.55 | 0 | 7.66 | 0 | 0.97 |

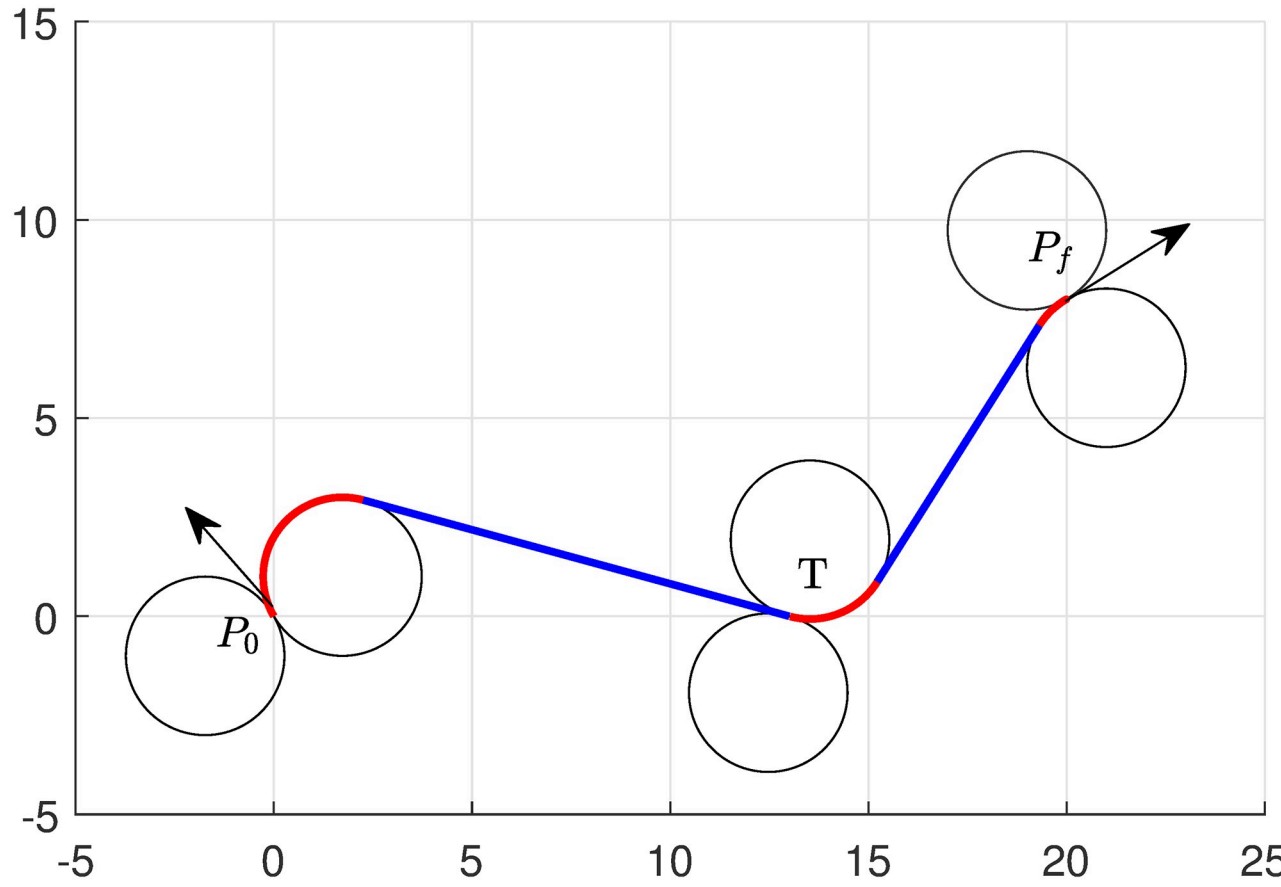

**Fig 3. Minimum path of UAV with single target touring of type RSLSR.**

We also solved Example 2 considering with smaller radii $r = 1$, and obtained the shortest path depicted in Fig 5. Optimal solutions for the arc lengths are included in Table 2. The total length obtained by the algorithm is 44.85, and the computational taken by the solver (Bonmin [39]) is 0.0625 seconds, the total time elapsed by the system 0.078 seconds, and the simulation time taken by MATLAB is 2 seconds.

We now compare our proposed model with the model given in [9, Section 5.1, Problem (Ps)]. In our comparison, we consider the single target touring problem-the problem description given in Example 3.

### Example 3

We recall now Example 1 with an initial angle $\pi/3$. Set starting point $P_0(0, 0, \pi/3)$ and destination point $P_f(20, 8, \pi/6)$. The task is given to UAV to tour target $T(13, 0)$ with a minimum path chosen from (15). It is assumed that the minimum radius of the turning circle for UAV is $r = 2$, and UAV also follows the same radius circle when it intercepts the target and moves towards its destination $P_f$.

Results obtained for both models (30) and [9, Section 5.1, Problem (Ps)] demonstrated in Figs 6 and 7. It is shown that *RSLSR* path is the optimum path to accomplish the task. In addition, subarc lengths of optimal solutions are provided in Table 3 for both models.

In our analysis, we observe that both models produce the similar kind of minimum path. We found that the total length of the path obtained by the models (30) and [9, Section 5.1,

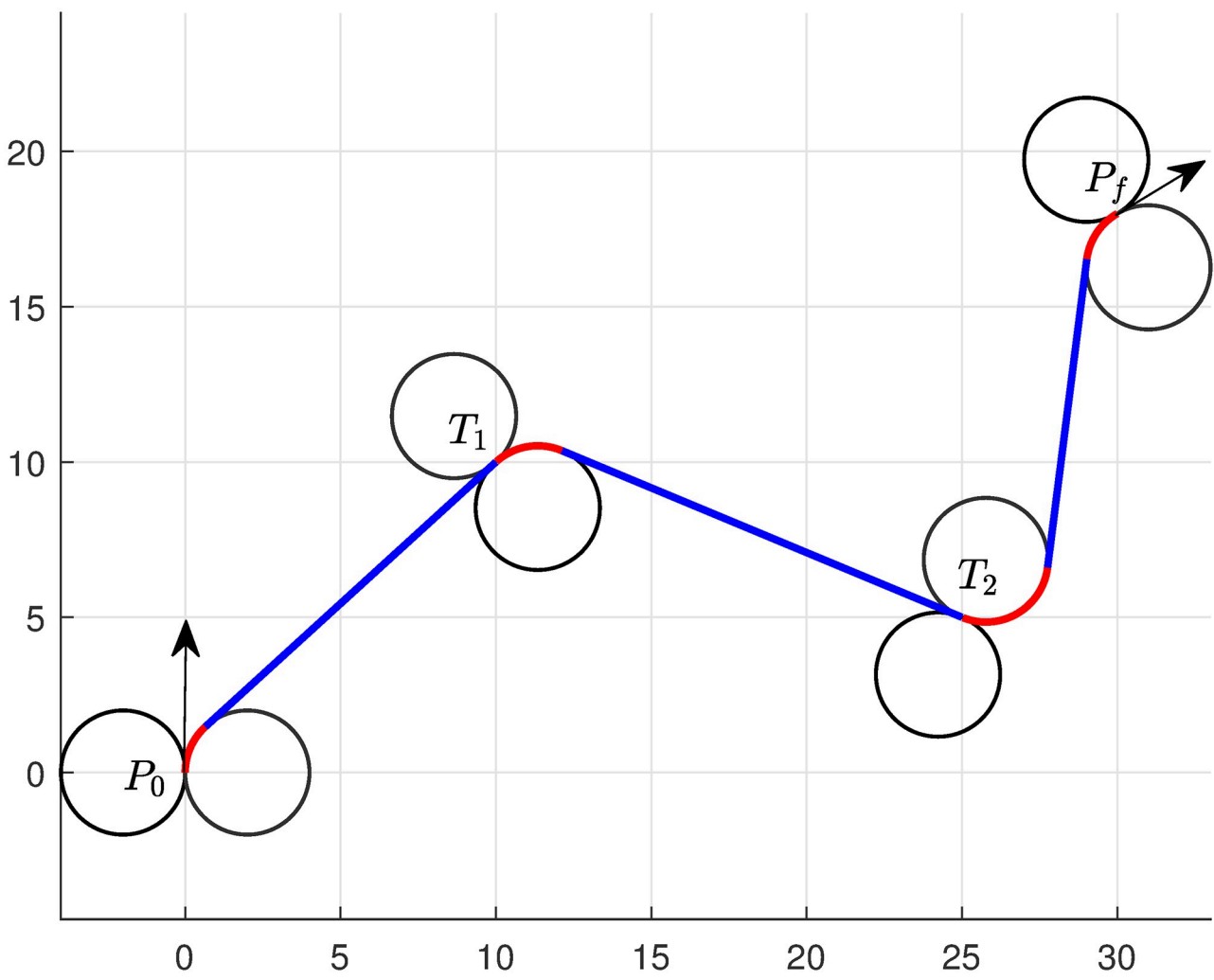

**Fig 4. Minimum path of UAV with two targets touring of type RSRSLSR.**

Problem (Ps)] is 24.30 and 24.13, respectively. Note that the computational time taken by the solver (Ipopt [40]) are 0.125 seconds and 0.135 seconds, and the total time elapsed by the system 0.155 seconds and 0.167 seconds, for models (30) and [9, Section 5.1, Problem (Ps)], respectively. In addition, the simulation time taken by MATLAB is approximately 1.5 seconds.

**Remark**: The proposed model (30) provides the optimal path of UAV, travels from starting point to destination with a target touring, is partitioned through 8 segments (circular(C) or straight line(S)). The concatenation of these segments is as follows. The lengths $L_{l_1}$ and $L_{l_2}$ (see Table 3) are considered for the segments of left-turn and right-turn respectively at the initial

**Table 2. Multiple targets touring—Numerical performance of model (30).**

| Targets/lengths | Left turn | Right turn | Straight line | Left turn | Right turn | Straight turn | Left turn | Right turn | Straight line | Left turn | Right turn |
|---|---|---|---|---|---|---|---|---|---|---|---|
| | $L_{l_1}$ | $L_{l_2}$ | $L_{l_3}$ | $L_{l_4}$ | $L_{l_5}$ | $L_{l_6}$ | $L_{l_7}$ | $L_{l_8}$ | $L_{l_6}$ | $L_{l_7}$ | $L_{l_8}$ |
| $T_1$ & $T_2$ $r = 2$ | 0 | 1.66 | 12.65 | 0 | 2.27 | 13.96 | 3.68 | 0 | 10.01 | 0 | 1.84 |
| $T_1$ & $T_2$ $r = 1$ | 0 | 0.81 | 13.42 | 0 | 1.12 | 14.90 | 1.66 | 0 | 12.16 | 0 | 0.78 |

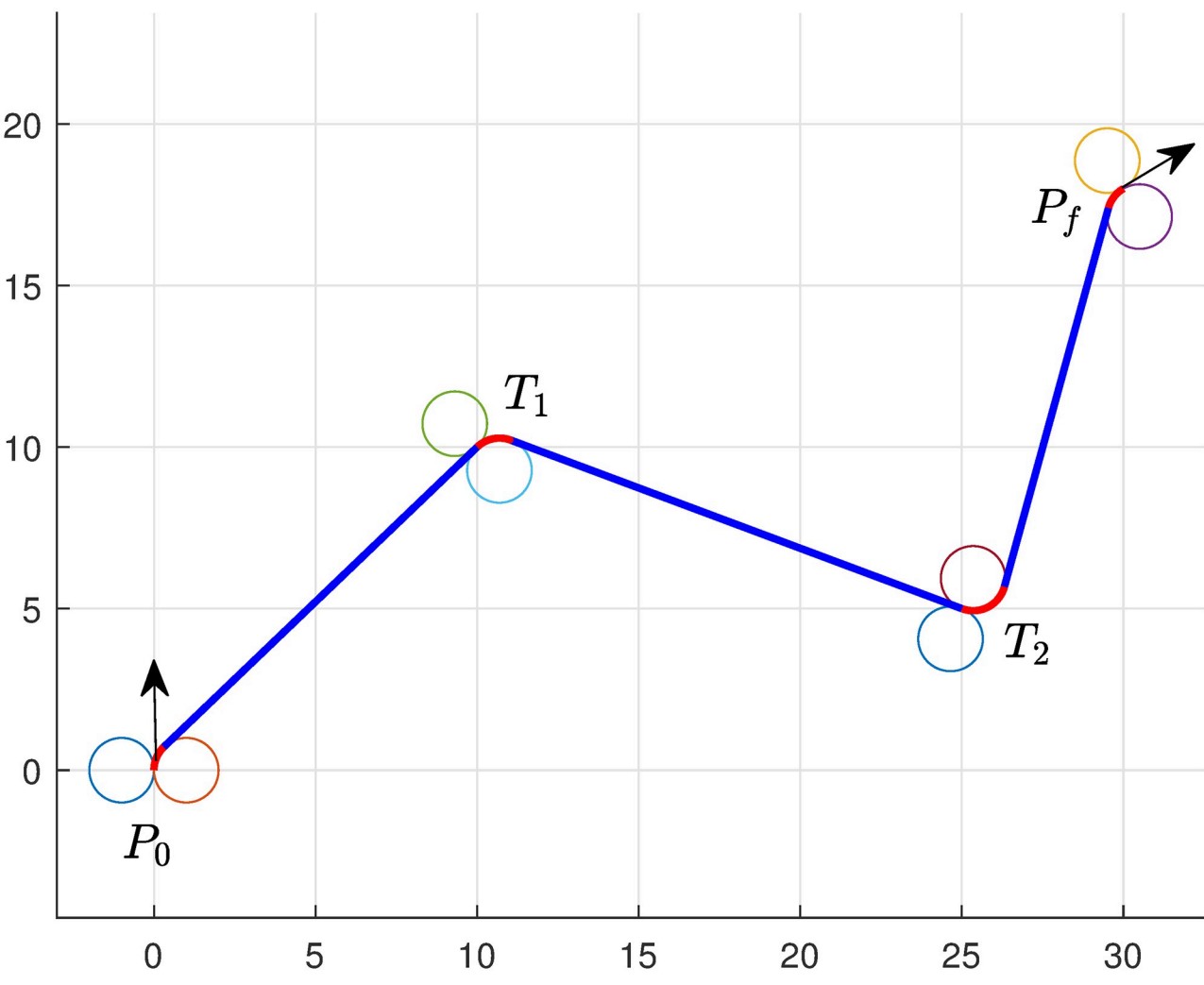

**Fig 5. Minimum path approximated for Example 2 when *r* = 1.**

point when the UAV approaches to target. The length $L_{l_3}$ for the straight line segment joining the terminal point of circular subarc $L_{l_1}$ or $L_{l_2}$ and the target point, and the length of the consecutive segment are $L_{l_4}$ and $L_{l_5}$ of the left-turn and right-turn respectively, when UAV is headed towards its destination from the target point; similarly, three more lengths ($L_{l_6}$, $L_{l_7}$ and $L_{l_8}$) of segments are required to reach the destination point. This is demonstrated in Table 3. On the other hand, the model [9, Section 5.1, Problem (Ps)] provides a concatenation of 10 segments with approximated lengths to obtain the optimal path for the same single target touring as mentioned above. In this case, the two segments $\hat{L}_{l_4}$ and $\hat{L}_{l_5}$ (see Table 3) of total 10 segments are additionally appeared. Moreover, for *n* targets touring, model [9, Section 5.1, Problem (Ps)] requires 5*n*+ 5 segments, whereas the proposed model (30) requires 3*n* + 5 segments with approximated lengths. As a result, there are need more ((5*n* + 5) − (3*n* + 5)) or 2*n* segments to calculate the optimal path for *n* targets touring in the model [9, Section 5.1, Problem (Ps)] than that of the segments in the model (30). This difference is reflected in our numerical experiments and we have verified this for a single target touring. The computational time taken by the solver (Ipopt [40]) are 0.125 seconds and 0.135 seconds, and the total time elapsed

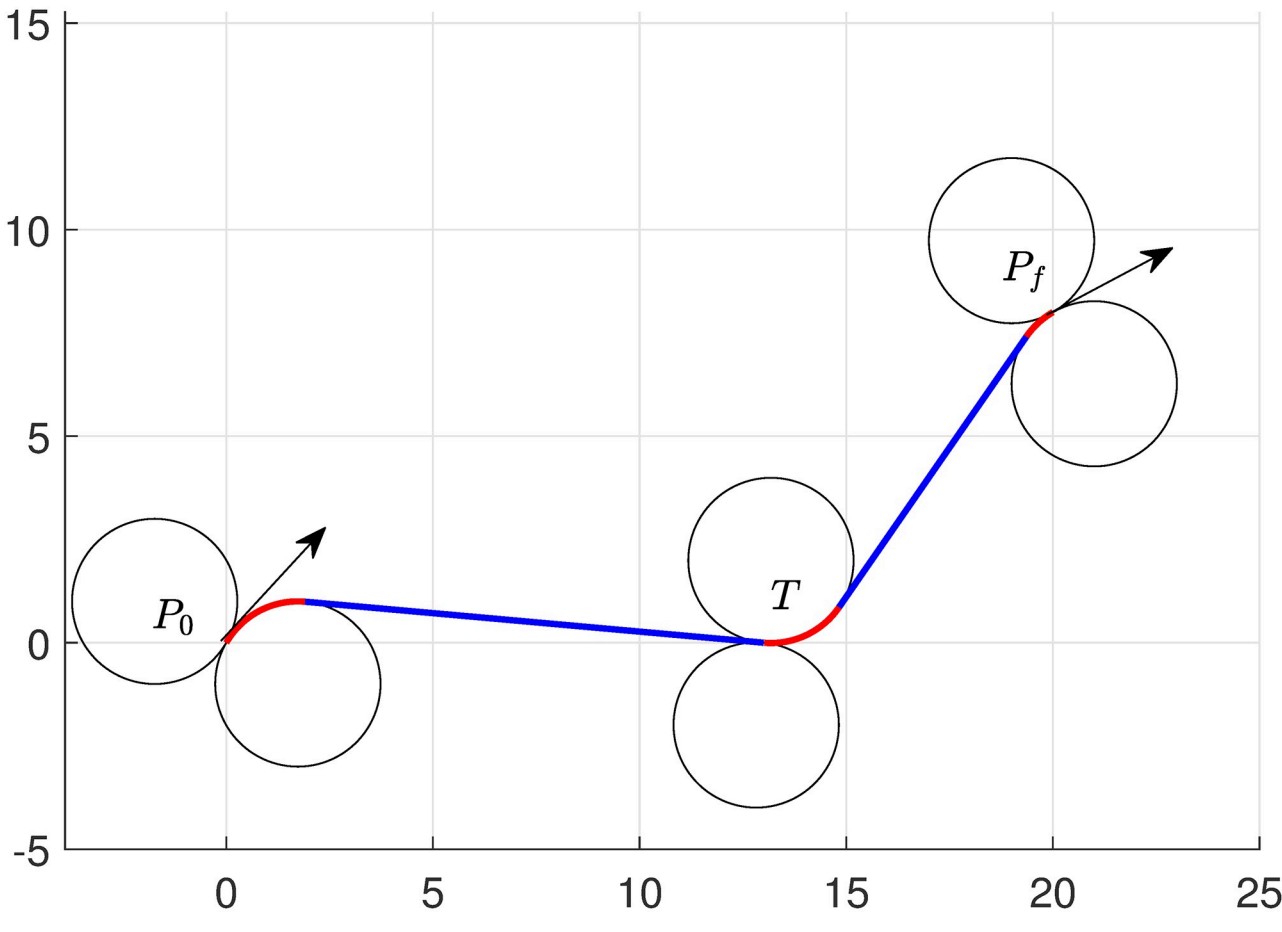

**Fig 6. Shortest path generated by the model (30).**

by the system 0.155 seconds and 0.167 seconds, for models (30) and [9, Section 5.1, Problem (Ps)], respectively. Therefore, the proposed model might have a computational advantage over the model stated in [9].

We have coded the problem in AMPL [41], and a number of solvers have been utilized, such as, Ipopt [40], SNOPT [42], Bonmin [39] and KNITRO [38] with default options to solve models (24) and (30). In our analysis, the computations have been performed on a Dell Inspiron laptop with 16 GB RAM and core i7 at 4.6 GHz. Each simulation of our analysis requires memory consumption of 29228 MB (combined RAM for AMPL and MATLAB during simulation) on average.

## 6 Conclusion

In this paper, *CSC* class path or subset thereof are exploited for optimal path planning of UAV under kinematic and target touring constraints. Numerical technique referred to as arc parameterization or switching time parameterization was characterized to find the minimum path for our proposed model. Furthermore, an efficient optimization algorithm was developed to enforce the optimality criterion. Extensive computational experiments were conducted to demonstrate the efficiency of the proposed model for single and multiple targets touring. AMPL was used to perform the tests, and a wide range of solvers have been utilized to solve

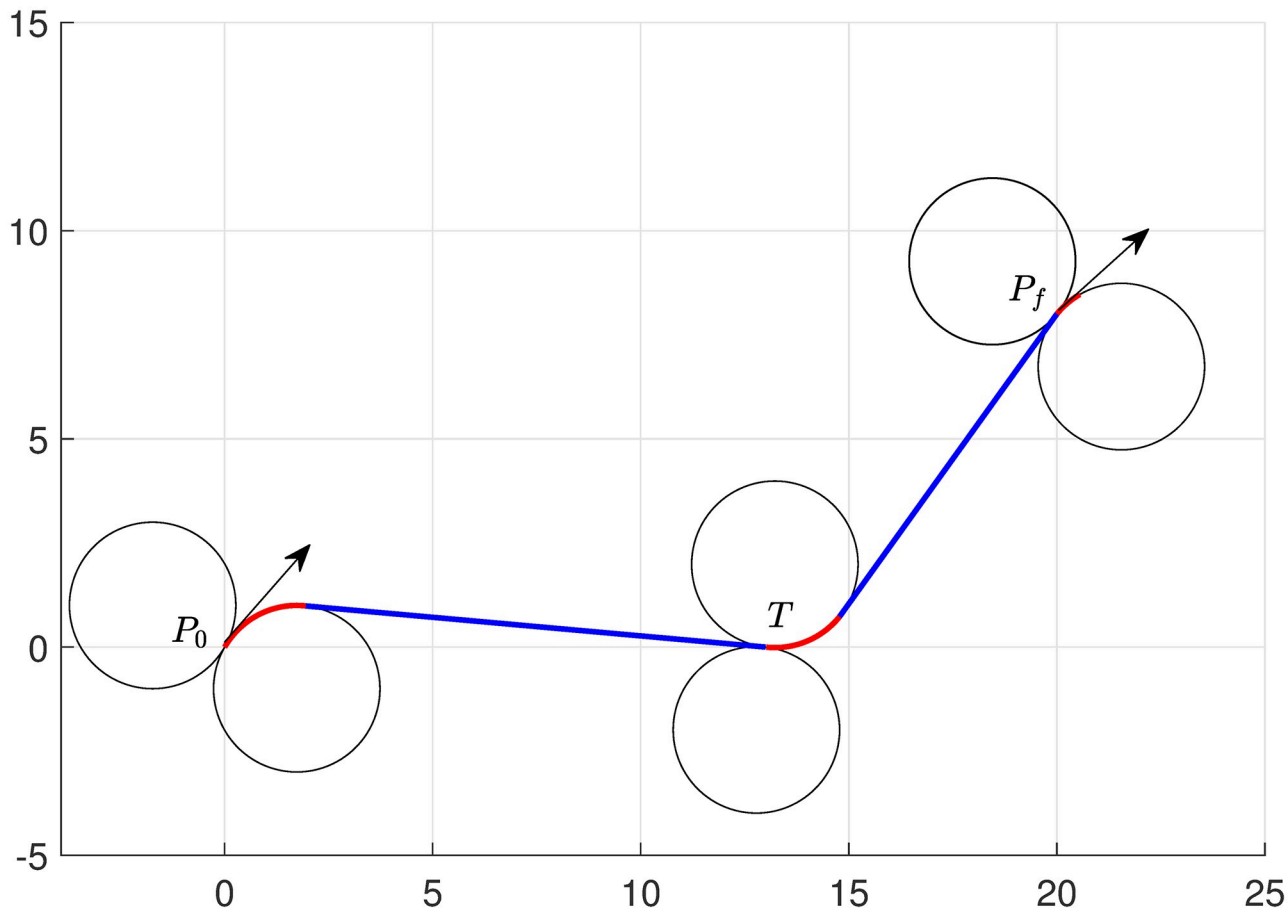

**Fig 7. Shortest path generated by the model [9, Section 5.1, Problem (Ps)].**

the problem. Also, we have simulated the optimal paths corresponding to each numerical experiment by using MATLAB.

The material is presented in this paper can be extended for the obstacle avoidance problem. UAVs fly through obstacles such as buildings, hills, restricted zones, etc., which intercept the normal flight paths of the UAV. Sensors monitoring the UAV's environment for fixed or moving obstacles are usually used for obstacle avoidance. The obstacle avoidance problem is closely associated with path planning because obstacles typically result in the re-planning of paths. For obstacle avoidance, our proposed model requires slight changes in the geometrical approach given in Fig 1 and then requires expressing the mathematical functions to construct the flying paths. For example, suppose the obstacle is circular, in that case, the target point $T$

**Table 3. Numerical performance of model (30) and [9, Section 5.1, Problem (Ps)].**

| Models | Left turn | Right turn | Straight line | Left turn | Right turn | Left Step | Right Step | Straight line | Left turn | Right turn |
|---|---|---|---|---|---|---|---|---|---|---|
|  | $L_{l_1}$ | $L_{l_2}$ | $L_{l_3}$ | $L_{l_4}$ | $L_{l_5}$ | $\hat{L}_{l_4}$ | $\hat{L}_{l_5}$ | $L_{l_6}$ | $L_{l_7}$ | $L_{l_8}$ |
| Problem (30) | 0 | 2.17 | 11.13 | 2.11 | 0 | . . . | . . . | 7.96 | 0 | 0.89 |
| Problem [9, (Ps)] | 0 | 2.32 | 10.13 | 1 | 0 | 1 | 0 | 8.95 | 0 | 0.73 |

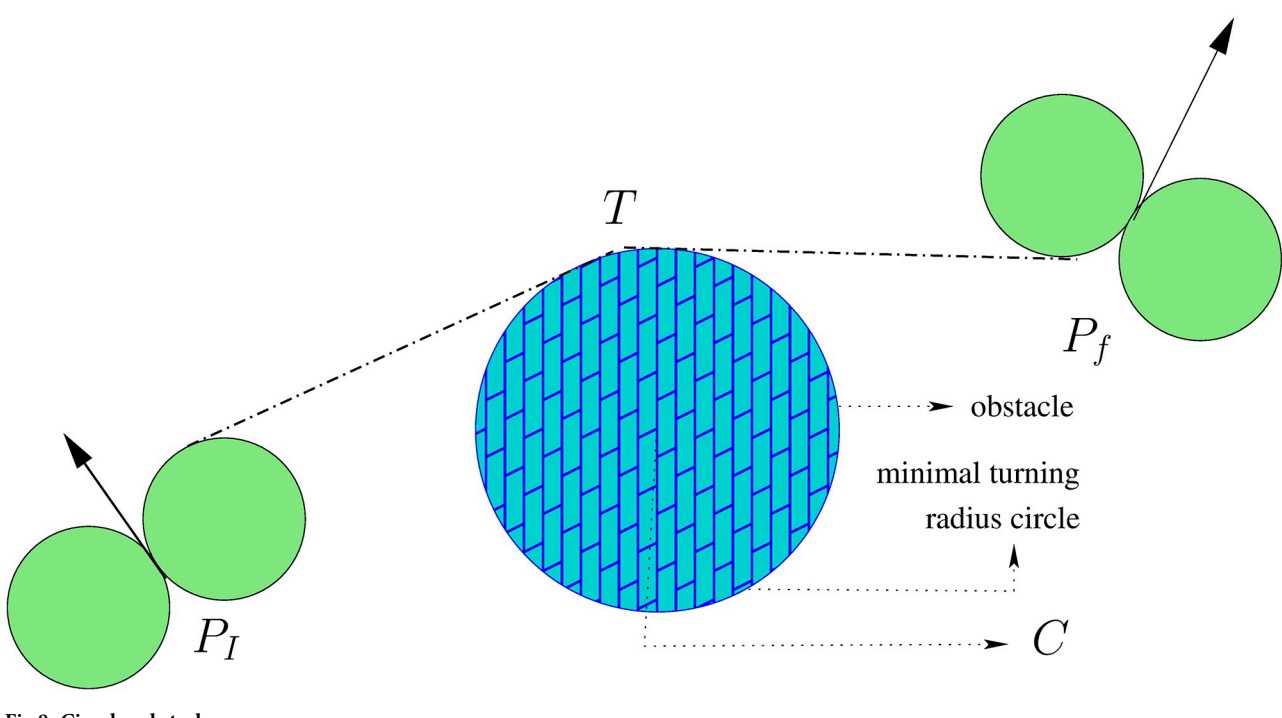

**Fig 8. Circular obstacles.**

must be on the circumference of the minimum turning circle centred at *C* (see Fig 8). The point *T* is variable here, and it is required to find *T* in such a way so that the flying path is minimum.

## Acknowledgments

The authors are very thankful to Dr Yalcin Kaya, Associate Professor, STEM, University of South Australia, Australia, for his suggestions and inspiration during this research.

## Author Contributions

**Conceptualization:** Mohammad Forkan, Mohammed Mustafa Rizvi.

**Formal analysis:** Mohammad Forkan, Mohammed Mustafa Rizvi.

**Investigation:** Mohammad Forkan, Mohammed Mustafa Rizvi.

**Methodology:** Mohammad Forkan, Mohammed Mustafa Rizvi.

**Resources:** Mohammad Forkan.

**Software:** Mohammed Mustafa Rizvi.

**Supervision:** Mohammed Mustafa Rizvi.

**Visualization:** Mohammed Mustafa Rizvi.

**Writing – original draft:** Mohammad Forkan.

**Writing – review & editing:** Mohammed Mustafa Rizvi, Mohammad Abul Mansur Chowdhury.

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
