## [Decision Letter · Decision Letter 0]

22 Jun 2022

PONE-D-22-12234Optimal Path Planning of Unmanned Aerial Vehicles (UAVs) for Targets Touring: Geometric and Arc Parameterization ApproachesPLOS ONE

Dear Dr. Rizvi,

Thank you for submitting your manuscript to PLOS ONE. After careful consideration, we feel that it has merit but does not fully meet PLOS ONE’s publication criteria as it currently stands. Therefore, we invite you to submit a revised version of the manuscript that addresses the points raised during the review process.

We look forward to receiving your revised manuscript.

Kind regards,

Hector Vazquez-Leal

Academic Editor

PLOS ONE

Journal Requirements:

[M. Forkan would like to express his gratitude to Ministry of

National Science & Technology, Bangladesh for providing financial help in the form of

NST fellowship with Reference no. 120005100-3821117, Reg. no. 9 & Session: 2020-2021.]

 [M. Forkan received grants from Ministry of National Science & Technology, Bangladesh for providing financial help in the form of NST fellowship with Reference no. 120005100-3821117, Reg. no. 9 & Session: 2020-2021.]

5. Please update your submission to use the PLOS LaTeX template. The template and more information on our requirements for LaTeX submissions can be found at http://journals.plos.org/plosone/s/latex.

Additional Editor Comments:

Dear Authors,

The optimal path planning for UAVs is an interesting problem. The authors develop an interesting proposal.

Please, consider the following comments:

1) Perform a comparison with other similar methods of literature

2) Report the computation time for the simulations

3) Report the memory consumption for each simulation

4) Please validate your proposal with a real robot.

5) Please, extend the explanation about how you will deal with obstacle avoidance in a future work, give more details.

Reviewers' comments:

Reviewer's Responses to Questions

**Comments to the Author**

1. Is the manuscript technically sound, and do the data support the conclusions?

Reviewer #1: Partly

Reviewer #2: Partly

2. Has the statistical analysis been performed appropriately and rigorously? 

Reviewer #1: No

Reviewer #2: I Don't Know

3. Have the authors made all data underlying the findings in their manuscript fully available?

Reviewer #1: Yes

Reviewer #2: Yes

4. Is the manuscript presented in an intelligible fashion and written in standard English?

Reviewer #1: Yes

Reviewer #2: Yes

5. Review Comments to the Author

Reviewer #1: This paper presents an optimal path planning model for UAVs to control their direction during target touring, where the UAV and target are at the same altitude. In this sense, the problem is performed on a 2-D space in which the orientation of the UAV is considered.

The paper is well structured, however, some sections need to be modified to improve the presentation and improve the content of this work:

1. Figure 1 and its description should be explained in a little more detail since the interpretation is confusing.

2. Equation 3 should be revised.

3. Equations 5, 6, 7, 8, and 9 are not written in a standard math text. Review notation for "sin, arg, cos, for all".

4. The authors can enrich the explanation of section 4 using algorithms or flow charts.

5. The results of section 5 should be contrasted with the results of similar works. In addition, it is essential that the authors place the computation time and the amount of memory spend to obtain each result.

Reviewer #2: The paper is interesting, and presents a congruent mathematical analysis. I consider that the use of the drone at the same height should be justified in a solid way, as this is a point of improvement to the proposed analysis. I also consider that more experiments with radii of circumferences smaller than 2 should be shown.

6. PLOS authors have the option to publish the peer review history of their article (what does this mean?). If published, this will include your full peer review and any attached files.

Reviewer #1: No

Reviewer #2: No

---

## [Author Response · Author response to Decision Letter 0]

16 Aug 2022

Regarding the concerns of the Editor and Reviewers, we made the changes in the manuscript and submitted two files ‘Manuscript’ (without track changes) and ‘Revised Manuscript’ with track changes. We would like to thank the Editor and Reviewers for their careful reading of the manuscript. Their comments have improved the presentation and text of our manuscript.

We have made a separate file to address the Editor and Reviewer's comments named ‘Response to Reviewers’ which has been provided.

---

## [Decision Letter · Decision Letter 1]

7 Sep 2022

PONE-D-22-12234R1Optimal Path Planning of Unmanned Aerial Vehicles (UAVs) for Targets Touring: Geometric and Arc Parameterization ApproachesPLOS ONE

Dear Dr. Rizvi,

Thank you for submitting your manuscript to PLOS ONE. After careful consideration, we feel that it has merit but does not fully meet PLOS ONE’s publication criteria as it currently stands. Therefore, we invite you to submit a revised version of the manuscript that addresses the points raised during the review process.

We look forward to receiving your revised manuscript.

Kind regards,

Hector Vazquez-Leal

Academic Editor

PLOS ONE

Additional Editor Comments (if provided):

The authors improved de manuscript. However, they must address the comments from the reviewer that ask for major revisions.

Reviewers' comments:

Reviewer's Responses to Questions

**Comments to the Author**

1. If the authors have adequately addressed your comments raised in a previous round of review and you feel that this manuscript is now acceptable for publication, you may indicate that here to bypass the “Comments to the Author” section, enter your conflict of interest statement in the “Confidential to Editor” section, and submit your "Accept" recommendation.

Reviewer #1: (No Response)

Reviewer #2: All comments have been addressed

2. Is the manuscript technically sound, and do the data support the conclusions?

Reviewer #1: Yes

Reviewer #2: Yes

3. Has the statistical analysis been performed appropriately and rigorously? 

Reviewer #1: Yes

Reviewer #2: Yes

4. Have the authors made all data underlying the findings in their manuscript fully available?

Reviewer #1: Yes

Reviewer #2: Yes

5. Is the manuscript presented in an intelligible fashion and written in standard English?

Reviewer #1: Yes

Reviewer #2: Yes

6. Review Comments to the Author

Reviewer #1: The document has substantially improved, however, some points mentioned above must be addressed and discussed in the respective sections.

1. Some mathematical expressions in section 3 are not numbered.

2. The comparison between the present work and similar ones is not clear, it is recommended to extend it in the discussion. Additionally, it is recommended to compare with more recent works less than 7 years old. If there are no recent works with which to compare, it is because the topic is not of interest, or what is the cause?

3. The authors report a memory consumption of 29228 MB, is this the data stored during the simulation in ROM (Hard Disk) memory?

Reviewer #2: no comment, I think it's ready

7. PLOS authors have the option to publish the peer review history of their article (what does this mean?). If published, this will include your full peer review and any attached files.

Reviewer #1: No

Reviewer #2: No

---

## [Author Response · Author response to Decision Letter 1]

18 Sep 2022

Dear Editor,

We submitted our revised manuscript with responses to the reviewer's comments. Provided the pdf version of three files ‘Manuscript’ (without track changes), ‘Revised Manuscript’ with track changes and 'Response to Reviewers'. We also provided a LateX file (only for manuscript) with all ps figures.

Thank you.

Kind regards,

Dr Mohammed Mustafa Rizvi

---

## [Editor Report · Decision Letter 2]

29 Sep 2022

Optimal Path Planning of Unmanned Aerial Vehicles (UAVs) for Targets Touring: Geometric and Arc Parameterization Approaches

PONE-D-22-12234R2

Dear Dr. Rizvi,

We’re pleased to inform you that your manuscript has been judged scientifically suitable for publication and will be formally accepted for publication once it meets all outstanding technical requirements.

Kind regards,

Hector Vazquez-Leal

Academic Editor

PLOS ONE

---

## [Editor Report · Acceptance letter]

4 Oct 2022

PONE-D-22-12234R2 

Optimal Path Planning of Unmanned Aerial Vehicles (UAVs)
for Targets Touring: Geometric and Arc Parameterization
Approaches 

Dear Dr. Rizvi:

I'm pleased to inform you that your manuscript has been deemed suitable for publication in PLOS ONE. Congratulations! Your manuscript is now with our production department. 

Kind regards, 

on behalf of

Dr. Hector Vazquez-Leal 

Academic Editor

PLOS ONE